# Hydrogen Sulfide: From a Toxic Molecule to a Key Molecule of Cell Life

**DOI:** 10.3390/antiox9070621

**Published:** 2020-07-15

**Authors:** Angeles Aroca, Cecilia Gotor, Diane C. Bassham, Luis C. Romero

**Affiliations:** 1Department of Genetics, Development and Cell Biology, Iowa State University, Ames, IA 50011, USA; bassham@iastate.edu; 2Institute of Plant Biochemistry and Photosynthesis, University of Seville and CSIC, 41092 Seville, Spain; gotor@ibvf.csic.es (C.G.); lromero@ibvf.csic.es (L.C.R.)

**Keywords:** Hydrogen sulfide, crosstalk, persulfidation, gasotransmitter, signaling molecules, human and plant therapies

## Abstract

Hydrogen sulfide (H_2_S) has always been considered toxic, but a huge number of articles published more recently showed the beneficial biochemical properties of its endogenous production throughout all *regna*. In this review, the participation of H_2_S in many physiological and pathological processes in animals is described, and its importance as a signaling molecule in plant systems is underlined from an evolutionary point of view. H_2_S quantification methods are summarized and persulfidation is described as the underlying mechanism of action in plants, animals and bacteria. This review aims to highlight the importance of its crosstalk with other signaling molecules and its fine regulation for the proper function of the cell and its survival.

## 1. Introduction

Hydrogen sulfide (H_2_S) is a flammable, colorless gas with a characteristic odor of rotten eggs. H_2_S naturally occurs in volcanic gases, natural gas and some well water and is also produced when bacteria break down organic matter in the absence of oxygen. H_2_S poisoning has mainly been observed in industrial settings. Thus, workers may be exposed to H_2_S in many industries, including agriculture, petroleum, and sewage processing [1]. H_2_S is toxic to humans and acute exposure to high amounts of H_2_S (>500 ppm) can lead to death [1]. The first reported biological experiment to study the effect of H_2_S in animals was published in 1908 and described the lethal effects of H_2_S gas when it was absorbed through the skin or directly administered to the stomach or rectum [2]. Since then, hundreds of articles reporting on the toxicological effects of H_2_S in various species, including humans, different cell types and organs, were published during the last century. This gas has always been considered toxic due to its ability to inhibit mitochondrial respiration through inhibition of cytochrome c oxidase, similar to hydrogen cyanide (HCN) [3]. Nevertheless, over the last few decades of the past century, several investigations were conducted on the presence of H_2_S as an endogenous product in bacteria and mammals. In the oral microbiota in humans, H_2_S was found to be responsible for oral halitosis and to be related to periodontal inflammation [4], and in the intestinal microbiota, H_2_S was found to be a component of flatus [5]. In parallel, several publications conducted detailed characterizations of the biochemical properties of the endogenous production pathways of H_2_S in different species. The mammalian enzymes responsible for H_2_S production are cystathionine beta-synthase, CBS; cystathionine gamma-lyase, CSE; and 3-mercaptopyruvate sulfurtransferase, 3-MST [6,7] (Figure 1), which have homologs in other species: in *Klebsiella pneumoniae*, CBS is the main source of H_2_S; in *E. coli*, 3-MST is the main source of H_2_S under aerobic conditions, while the cysteine desulfurase (IscS) is the primary source under anaerobic conditions [8].; Similar to mammals, *C. elegans* has three H_2_S-synthesizing enzymes, CSE, CBS and 3-MST [9,10,11]. In plants, H_2_S production was first observed in 1964, when the release of sulfide components in Liliaceous vegetables was described [12], and then, in 1968, Cormis described the emission of H_2_S from plants after exposure to SO_2_ [13]. In 1987, two enzymes, L- and D-cysteine desulfhydrases, were found to catalyze the production of H_2_S in chloroplasts and mitochondria [14] (Figure 1). In recent years, several characterizations of these enzymes have been conducted, revealing detailed H_2_S biosynthesis and sulfur assimilation pathways in plants [15,16].

## 2. Physiological Role

During the past century, H_2_S was thought to only be a toxic molecule, and it was not until 1990 that Kimura and coworkers revealed its role in essential functions in human physiology, opening a new emerging field in life science [17]. The first physiological assay published in 1996 demonstrated that H_2_S acts as an endogenous neuromodulator [18]. The participation of H_2_S in many physiological and pathological processes in animals has been described over the last two decades, including its role in the regulation of cell proliferation, apoptosis, inflammatory processes, hypoxia, neuromodulation, and cardioprotection [19,20,21]. Therefore, H_2_S is now accepted as playing roles as a gasotransmitter (gaseous signaling compound) that is as important as nitric oxide (NO) and carbon monoxide (CO) in mammals, and as a signaling molecule that is as important as hydrogen peroxide (H_2_O_2_) in plants [18,22,23]. The term ‘gasotransmitter’ was introduced in 2002 to describe these molecules, which share common characteristics: they are endogenously produced, with a signaling role, generated by enzymatic pathways, and permeable to cell membranes; their endogenous biosynthesis may be regulated; and their effect is dose-dependent (Figure 1). The scientific interest in H_2_S in the past was mainly due to its role in important and devastating human diseases, such as neurodegenerative disorders, including Alzheimer’s disease, Parkinson’s disease, and vascular dementia [24,25,26]; Huntington’s disease [27]; and cancer [28,29,30].

Although the first descriptions of the effects of H_2_S in plants were from the 1960s [31], interest in the role of H_2_S in plant systems arose later. It was not until the past decade that the effects of H_2_S were described in seed germination [32], the number and length of adventitious roots [33] and the regulation of genes involved in photosynthesis [34]. Thereafter, the protective effects of exogenous H_2_S against different stresses were documented, such as protection against oxidative and metal stresses [32,35,36,37,38,39,40], drought and heat tolerance [39,41], and osmotic and saline stresses [42]. Thus, publications on these dose-dependent effects of H_2_S have emerged, postulating H_2_S to be an important signaling molecule that has analogous functions in plant systems to those previously described in mammals. H_2_S was also shown to be a regulator of other important physiological processes in plants, such as stomatal closure/aperture [43,44,45,46]; thus, its importance in drought stress relief is due to the ability of H_2_S to induce stomatal closure in *Arabidopsis thaliana* [46,47]. Another positive effect of H_2_S was described by Dooley et al., who showed that low doses of H_2_S strongly affected plant metabolism, improved germination, caused significant plant growth and increased biomass, leading to a higher fruit yield [48]. H_2_S has also been shown to be involved in the regulation of flower senescence in plants [49], in lateral root formation in tomato mediated by auxin-regulation [50] and in nicotine biosynthesis in tobacco [51].

More recently, there has been increasing interest in the effect of H_2_S on autophagy regulation in the scientific community. In mammals, the protective effect of H_2_S against some of the diseases mentioned above has been linked with the regulation of autophagy [52,53]. Autophagy is a cellular catabolic pathway that is evolutionarily conserved from yeast to mammals, and it involves the digestion of cell contents to recycle nutrients or to degrade damaged or toxic components. The AMP-dependent kinase (AMPK) and mammalian target of rapamycin (mTOR) pathways play important roles in the control of autophagy. To this end, activation of AMPK or inhibition of mTOR has been shown to activate autophagy [54]. Exposure to H_2_S has been shown to cause a significant increase in AMPK phosphorylation, which increases its activity and inhibits the activation of downstream targets, such as mTOR [55]. In plants, H_2_S was shown to inhibit autophagy by preventing ATG8 (autophagy-related ubiquitin-like protein) accumulation [56]. H_2_S is able to inhibit starvation-induced autophagy in Arabidopsis roots, and this repression is independent of redox conditions [57].

The first mechanism proposed for H_2_S was based on its chemical properties, since this nucleophilic molecule is able to react with reactive oxygen/nitrogen species and reduce the cellular oxidative state [58,59]. In addition, H_2_S is able to regulate several antioxidant enzymes, such as ascorbate peroxidase (APX) [60,61,62], catalase (CAT) [63,64], superoxide dismutase (SOD) [63,65], and glutathione reductase (GR) [62], and non-enzymatic compounds, such as the glutathione anti-oxidant pool [66,67].

The antioxidant role of H_2_S has been the focus of numerous studies in mammalian systems as a critical mediator of multiple pathophysiological processes [68]. In plants, the number of studies on the effects of H_2_S in the model plant Arabidopsis has increased in recent years; in addition, the effects of H_2_S in agricultural crops are relevant as an exogenous treatment to cope with economic loss due to environmental stress. The effects of exogenous (pre-)treatment with water-soluble donors of H_2_S have been the focus of numerous studies in several agricultural species. Fotopoulos et al. have reviewed these studies regarding the effects of H_2_S on plant growth, its ability to improve resistance against abiotic and biotic stress, and its positive postharvest effects [69]. Thus, a better understanding of the mechanism of action of H_2_S is important to fight against crop loss. This knowledge would help in agricultural sustainability and in producing the food required by the increasing world population [70].

## 3. Quantification Methods of H_2_S

H_2_S in aqueous solution can be found as hydrogen sulfide gas (H_2_S) or in one of its dissociated forms, hydrosulfide anions (HS^−^) and sulfide anions (S^2−^), although at physiological pH, S^2−^ is only found in a negligible concentration. In addition, H_2_S can bind to some biological matrixes (proteins, glutathione, etc.) and can dissociate in response to a physiological stimulus into free H_2_S. Moreover, the anion HS^−^ and H_2_S have a high propensity to oxidize, especially in the presence of trace metal ions and oxygen in water solutions [71]. Therefore, accurately and reliably measuring H_2_S in vivo is an arduous task. Solutions should be prepared under a nitrogen or argon atmosphere, and H_2_S volatilization should be prevented using septum-sealed vials.

**Methylene blue method:** The first quantification method of H_2_S was published in 1949 and included spectrophotometric determination using the methylene blue method, but the sensitivity was very low [72]. In 1969, an improved method was described based on the methylene blue method using *n,n*-dimethyl-p-phenylenediamine sulfate, which increased the sensitivity of the method by 10%, with concentration limits of 1 to 1000 µM [73]. However, the main drawbacks of this method for biological sample measurements are its low sensitivity, overestimation of H_2_S under acidic pH and interference due to the turbidity of biological samples.

**Monobromobimane derivatization (MBB):** this is another method that is extensively used in biological samples to measure the H_2_S concentration. In this method, H_2_S is derivatized into a sulfide-dibimane product that can be subsequently measured by HPLC due to its fluorescence [74]. The sensitivity of MBB reaches the nanomolar range, but the weaknesses of this method are the instability of the standards and the need for exhaustive control of pH when comparing samples [75]. Another advantage of this method is that it allows the detection and quantification of all three sulfide biological forms: free hydrogen sulfide, acid-labile sulfide and bound sulfane sulfur [76]. An improved method based on MBB was developed that included ^36^S-labeled sulfide- dibimane as an internal standard and measured derivatized products by liquid chromatography-mass spectrometry [77]. This improvement ensured the sensitivity and feasibility of the method and also made it suitable for large-scale analysis.

The gas chromatography method was first used in 1988 [78] and was recently improved using a chemiluminescence sulfur detector to increase its sensitivity up to 0.5 pmol [79]. Nevertheless, the need for special equipment and special care in sample processing make this method not widely used.

**Specific ion electrodes and polarographic sensors** are others tools that have been used for H_2_S quantitation. Their easy handling and the relatively inexpensive cost of electrodes make this method a good choice for several studies. Since this method does not require derivatization, it can be used for real-time measurements in biological samples. Nevertheless, it was reported that the sensitivity of this method is apparently affected by the materials used in the electrodes [75], and the main disadvantage of this method is the difficulty of calibration due to the formation of Ag_2_S on the electrode surface.

**H_2_S-selective fluorescent probes** are receiving increasing interest since they are powerful tools for the detection and quantification of H_2_S in biological samples. Fluorescent probes have very high sensitivity and can be used for real-time measurements even in specific tissues or cellular compartments. The reactivity and specificity of these probes are based on the characteristic nucleophilicity of the anion HS^−^, on the reduction of an azide by H_2_S into an amine compound, or on the quenching effect of Cu^2+^ of a nearby fluorophore and the strong affinity of anion sulfide for this heavy metal ion [80]. In addition, the increasing interest in other sulfane sulfur molecules, such as persulfides (R–S–SH), polysulfides (R–S–S_x_–S–R), and hydrogen polysulfides (H_2_S_x_), in biological systems, has led to the development of probes that are able to detect these species. Several recent reviews summarize the latest probes reported in the literature [80,81]. However, the main limitation of these probes is the irreversible reaction that removes H_2_S from the pool, leading to saturation [71]. Motivated by these limitations, Takano et al. designed a reversible fluorescent probe to detect these sulfane sulfur species, SSip-1, based on the ability of sulfane sulfur to bind other sulfur atoms [82]. In parallel, Dulac et al. developed a new method of reversible quantification in biological fluids, detecting up to 200 nM of H_2_S at pH 7.4 using hemoglobin I from *Lucina pectinate* and a fluorophore [83]. Therefore, these reversible fluorescent probes allow the dynamics of intracellular sulfane sulfur molecules to be studied. Since the cytotoxicity of these probes is a drawback for their use in in vivo studies, a new low-cytotoxic biosensor for H_2_S has recently been developed based on anthracene derivatives [84]. A further disadvantage in the plant system is the inability of these probes to penetrate the cell wall. Although there are few reports on the use of fluorescent probes for H_2_S detection in plants, selective detection of intracellular H_2_S was described using the probe WSP-1 [3′-methoxy-3-oxo-3H-spiro [isobenzofuran-1, 9′-xanthen]-6′-yl 2-(pyridin-2-yldisulfanyl) benzoate] in tomato roots [85]. However, the use of fluorescent probes for quantification of H_2_S also has some limitations, such as a limited sensitivity range, interference by tissue autofluorescence when the emission/excitation wavelengths are similar, or non-specific reactions with other biological thiols. Thus, the fluorescent probe or the H_2_S quantification method must be carefully chosen by the researcher depending on the sample, the tissue or organelle target, the sensitivity range, the equipment and the budget.

Data from in vivo measurements have been in conflict for a long time. The first data reported used the methylene blue method, and the H_2_S concentration was above 35 µM and from 50 to 160 µM in mammalian plasma and brains, respectively [86,87]. However, these measurements were incorrect and using newly described methods, the H_2_S concentration was more accurately reported at approximately 0.7–3 µM in mammalian plasma [74]. Moreover, it has been recently reported than H_2_S concentration in mouse plasma is about 15 nM [88]. Taken altogether, the H_2_S concentration has been estimated in the range of nanomolar, but the H_2_S concentration may depend on the microenvironment at the precise moment of the H_2_S measurement. For example, H_2_S can be bound to sulfur compounds or conjugates and H_2_S released under a certain stress or stimulus [89], H_2_S biosynthesis may be up- or downregulated in particular scenarios [90,91], and H_2_S consumption or accumulation may be modulated by the regulation of H_2_S detoxification enzymes [92]. In addition, different H_2_S concentrations have been reported in different tissues, species or stages of life. In cucurbit plants, the H_2_S concentration was reported to be higher in leaves from older plants [93], while in Arabidopsis, the highest H_2_S concentration was found in 2-week-old seedlings and gradually decreased in 4–10-week-old plants [94]. The highest H_2_S content in Arabidopsis was found in flowers, while the lowest H_2_S concentration was found in cauline leaves [94]. Determination of the H_2_S concentration is further complicated by the fact that some stimuli only affect the H_2_S concentration in some tissues, such as nicotine, which affects the H_2_S concentration in the mouse kidney and heart but not in the brain and liver tissues [95]. A further important consideration must be taken when quantifying H_2_S in biological samples, since H_2_S is spontaneously oxidized in the presence of molecular O_2_ [96]. Therefore, the method of quantification must be accurate, and sample management must be taken into account because the H_2_S concentration may not remain constant under certain conditions.

## 4. H_2_S Mechanism of Action

The underlying mechanisms of H_2_S action are poorly understood. There is an important effect of H2S binding to heme moieties in target proteins such as cytochrome c oxidase, hemoglobin and myoglobin, among others [97]. It has, however, become widely accepted that a huge number of the processes controlled by H_2_S are caused by a posttranslational modification of cysteine residues called persulfidation [98,99,100]. Protein persulfidation is an oxidative posttranslational modification of cysteine residues caused by H_2_S, in which a thiol group (–SH) is transformed to a persulfide group (–SSH). Sulfane sulfur species, persulfides and polysulfides are more nucleophilic than H_2_S and therefore more effective at persulfidation [101]. Due to the intrinsic instability of persulfides and their higher reactivity than thiols, protein persulfides largely remain understudied. Nevertheless, over the last decade, study of this protein modification has become more relevant for researchers because it can affect protein function, localization inside cells, stability, and resistance to oxidative stress [23,60,102,103,104]. The broad physiological importance of persulfidation has only recently started to emerge; a proteomic analysis revealed that approximately 10–25% of liver proteins contain this modification [104], and at least 5–10% of the entire proteome may undergo persulfidation in plants [105]. Several detection methods have been developed in recent years based on the nucleophilic characteristic of persulfide groups. Conversely, due to their instability and similarity to thiol groups, the development of a specific method for persulfide detection has become a challenge. These detection methods have recently been reviewed, including further explanations of the reactions and procedures [106,107,108]. Using these methods, researchers have been able to decipher the mechanism of action of H_2_S through persulfidation in several diseases, such as cancer, neuronal degeneration diseases, or ischemia–reperfusion injury. Persulfidation of the α subunit of ATP synthase [109], lactate dehydrogenase A [110], the K_ATP_ channel [99,111], and MEK1 [112] contributes to cancer promotion. In Parkinson’s disease patients, the E3 ubiquitin ligase PARKIN shows a decrease in persulfidation, which decreases its enzymatic activity [113]. Keap1 persulfidation protects gastric epithelial cells from ischemia/reperfusion injury [114]. In mammals, the mechanism of action of H_2_S has been deeply studied since 2009, when Mustafa et al. described this new posttranslational modification [104]. By contrast, in the plant system, persulfidation has been described more recently [60], but a greater number of proteins have been shown to undergo this modification [105]. A total of 3147 proteins were found to be persulfidated in Arabidopsis leaves under physiological conditions, suggesting that this number may be higher under certain stress conditions [105]. These proteins are mainly involved in important biological pathways, such as the tricarboxylic acid cycle, glycolysis, Calvin cycle, photorespiration and autophagy. Further physiological studies of these proteins must be performed to decipher the role of persulfidation in these biological pathways. Nevertheless, initial studies in plants demonstrated that persulfidation regulates the enzymatic activity of chloroplastic glutamine synthetase (GS2), cytosolic ascorbate peroxidase (APX1), and cytosolic glyceraldehyde 3-phosphate dehydrogenase (GapC1) [60]. Persulfidation regulates the cytosolic/nuclear localization of GapC1, allowing it to likely act as a transcription factor [102]. The actin cytoskeleton and root hair growth are regulated through persulfidation of ACTIN2 [115]. Furthermore, ethylene biosynthesis is regulated by persulfidation of 1-aminocyclopropane-1-carboxylic acid oxidase (ACO1) in tomato [116]. Recently, a peroxisomal proteome study in Arabidopsis revealed that the interplay of different PTMs such as *s*-nitrosation, nitration, persulfidation, and acetylation regulates redox signaling to protect proteins against oxidative damage [117]. From an evolutionary point of view, it is reasonable to assume that ancestral purple and green sulfur bacteria lived in an H_2_S-rich atmosphere; and therefore bacteria developed H_2_S-mediated signaling processes to resist oxidative stress. Similarly as peroxide (H_2_O_2_) produces ROS, persulfide (H_2_S_2_) produces RSS (reactive sulfide species), but with the difference that persulfides can be produced with several sulfur molecules (S_x_) and stored [118].

This assumption was described in a recent study, in which a proteomic evaluation of *Staphylococcus aureus* showed that many proteins regulated by persulfidation were involved in reactive oxygen and nitrogen species (ROS and RNS) stress-responses and that bacterial virulence was regulated by persulfidation of the HTH-type transcriptional regulator MgrA (MgrA), a global virulence regulator [119].

All these data suggest that persulfidation is a conserved mechanism of H_2_S signaling throughout all kingdoms of life.

## 5. Crosstalk of H_2_S with Other Signaling Molecules

### 5.1. Nitric Oxide

It is well established that H_2_S regulates different physiological processes in cells directly or by crosstalk with other signaling molecules. There is clear evidence of crosstalk of H_2_S and NO in the literature. In mammals, both gasotransmitters interact with each other to modulate the cardiovascular system by regulating angiogenesis and endothelium-dependent vasorelaxation [120,121], and to modulate Alzheimer’s disease by regulating pathways involved in the central nervous system [122]. Furthermore, inhibition of NO generation by H_2_S has been extensively studied [123,124,125], but there is also evidence that NO can activate the production of H_2_S in endothelial cells [126]. However, NO can bind to cystathionine β-synthase (CBS), which is responsible for H_2_S biosynthesis and can impede its enzymatic activity [127], showing the complexity of the crosstalk between these two gasotransmitters.

In plants, NO and H_2_S play crucial roles in the regulation of multiple responses towards a variety of abiotic and biotic stresses [44,128,129], including stomatal closure/aperture [46], modulation of photosynthesis [34,130,131] and autophagy [56,132,133]. NO levels increase in plants under drought stress, which helps plants mitigate the negative effects of water deficit. NO increases the activities of antioxidant enzymes such as catalase (CAT), superoxide dismutase (SOD), ascorbate peroxidase (APX), glutathione reductase (GPX), and peroxidase (POD), and NO is an important player in ABA-induced stomatal closure, minimizing plant transpiration [134]. H_2_S application reduces the accumulation of NO in guard cells, causing stomatal opening in the presence of light and preventing stomatal closure in the dark [45]. Exogenous H_2_S induces stomatal closure through the regulation of ATP-binding cassette (ABC) transporters, while scavenging H_2_S can partially block ABA-dependent stomatal closure, indicating the protective role of H_2_S in plants against drought stress [46]. It has been demonstrated that H_2_S acts downstream of ABA and upstream of NO [44]. H_2_S-induced stomatal closure can be reversed by cPTIO (a NO-specific scavenger), confirming that the function of H_2_S in stomatal closure is mediated by NO [44]. It has been proposed that these contradictory effects of H_2_S on stomatal movement and its crosstalk with NO depend on the environment or the age of the plant [135]. In agricultural crops such as pepper plants, the crosstalk between NO and H_2_S plays an important role in the tolerance to iron deficiency and salt stress [136]; and these gasotransmitters partially modulate the NADPH-generating system by regulating 6-phosphogluconate dehydrogenase (6PGDH) and NADP-malic enzyme (NADP-ME) [137].

Furthermore, NO and H_2_S are involved in the functional regulation of proteins, frequently with opposite effects. *s*-nitrosation of GAPC abolishes its catalytic activity, whereas persulfidation increases its activity [60,138]. Nonetheless, a cooperative effect of both signaling molecules can also be observed in the case of cytosolic ascorbate peroxidase (APX1), which can be S-nitrosylated by NO or persulfidated by H_2_S, both of which increase the activity of the enzyme. These reversible modifications may protect the enzyme from irreversible oxidation under abiotic stress, where the oxidative stress increases and s-nitrosothiols have usually been observed [139].

H_2_S and NO may chemically interact and produce novel reactive molecules, such as nitroxyl (HNO) and nitrosothiols (RSNO) [140,141], which have their own outcomes. Recent studies also demonstrated that persulfides can produce NO using nitrite via intermediates such as polysulfide, SNO^−^ (thionitrite) and S_2_NO^−^ (perthionitrite, nitrosodisulfide) [142,143,144]. Therefore, H_2_S and NO interaction forms some intermediates, which also have significant roles in cell signaling.

### 5.2. Carbon Monoxide

Carbon monoxide is another important gasotransmitter in animals; carbon monoxide is generated from oxidative degradation of heme by the enzyme heme oxygenase. CO may inhibit CBS activity and therefore may modulate H_2_S biosynthesis [145]. Exogenous H_2_S also upregulates the CO/heme oxygenase system in the pulmonary arteries of hypoxic rats [146] and stimulates heme oxygenase levels in mouse retinal ganglion cells (RGC-5 cells) [147]. In plants, auxin induces endogenous H_2_S and CO during the initiation of lateral root primordia, and this growth is promoted by H_2_S but depends on the endogenous production of CO [50,148]. Furthermore, in a similar way as in mammals, exogenous H_2_S induces an increase in the transcription of heme oxygenase and its activity in tomato and cucumber roots [149,150]. Therefore, it has been suggested that H_2_S regulates the feedback loop between the CO/heme oxygenase system and auxin during lateral root initiation [151]. Although the crosstalk between H_2_S and CO needs further study, previous research on plant and animal systems provides evidence for an interrelationship of these two signaling molecules.

### 5.3. Hydrogen Peroxide

Hydrogen peroxide (H_2_O_2_) is a well-known signaling molecule in plants. H_2_O_2_ emerged as a key signaling molecule that enhances abiotic stress resistance by modulating the expression of resistance genes and antioxidant enzyme activities. Recently, signaling crosstalk between NO and H_2_S with H_2_O_2_ has been shown to induce thermotolerance in maize seedlings [152]. Hydrogen peroxide is also involved in H_2_S-induced lateral root formation in tomato seedlings, revealing that the cell cycle regulatory genes modulated by H_2_S, such as up-regulation of *SlCYCA2;1*, *SlCYCA3;1*, and *SlCDKA1*, and down-regulation of *SlKRP2*, are prevented by co-treatment with H_2_O_2_ scavengers [153]. A study in *Vicia faba* revealed crosstalk between H_2_S and H_2_O_2_ in salt stress-induced stomatal closure, with H_2_S being downstream of H_2_O_2_ [154]. This observation was also described in white clover, where H_2_S acts as a downstream signal of H_2_O_2_ and NO in response to dehydration [155]. A recent study showed a newly discovered crosstalk between H_2_S and H_2_O_2_ in another abiotic stress response, in which H_2_S can act as a positive regulator of Vacuolar H + -ATPase, while H_2_O_2_ acts as a negative regulator during cadmium stress in cucumber roots [156]. In mammalian cells, H_2_O_2_ is a key signal in redox regulation, and as it occurs in plants, these regulatory pathways may also be influenced by H_2_S. H_2_O_2_ is produced by NAPDH oxidases in the plasma membrane and is transported to the cytosol through protein channels named aquaporins (AQP3, AQP8, and AQP9). It has been demonstrated that treatment with H_2_S is sufficient to block H_2_O_2_ cell permeability in unstressed cells, and this phenomenon is mediated by the persulfidation of cysteine 53 of AQP8 [157]. By contrast, H_2_S production is dependent on the levels of H_2_O_2_ produced by NADPH oxidase, which attenuates the phosphorylation of vascular endothelial growth factor receptor 2 (VEGFR2) [158]. As with NO and CO, autoregulation of these signals may be influenced by their generation, increasing or reducing their intracellular concentrations depending on the levels of each other. Another example was provided by Feng et al., who found that autophagy was induced by H_2_O_2_ through ER stress in cardiac fibroblast cells and that H_2_S was able to suppress autophagic flux by inhibiting ROS production and preserving mitochondrial function [159]. All these studies establish a link for H_2_S/H_2_O_2_ crosstalk.

### 5.4. Hormones

H_2_S is a regulator of glucose homeostasis and plays an important role in the metabolism of hormones, such as insulin and glucagon [160,161]. It has been demonstrated that β cells of the pancreas can produce high levels of H_2_S, predominantly by cystathionine *γ*-lyase (CSE), which blocks glucose-stimulated insulin secretion [162]. This effect is caused by increased endoplasmic reticulum stress, leading to apoptosis of β cells, which drives the reduction in insulin secretion [161]. Some other studies revealed the importance of H_2_S in the modulation of estrogen receptor expression and its anti-proliferative effect on vascular smooth muscle cell growth and proliferation [163]. Further research concluded that the antiatherosclerotic effect of estrogen is mediated by CSE-generated H_2_S and that H_2_S production in the liver and vascular tissues is enhanced by estrogen via its stimulatory effect on CSE activity [164]. In a recent study, H_2_S signaling was also linked with the regulation of two endocrine hormones associated with longevity control, growth hormone and thyroid hormone. Thyroid hormone suppresses H_2_S production by inhibiting CSE gene expression, while growth hormone controls its substrate availability via autophagy. Surprisingly, CSE-generated H_2_S is necessary for the feedback regulation of thyroid and growth hormones [165].

Moreover, H_2_S has been linked to plant hormone signaling, such as gibberellin (GA) [166], auxin [33], jasmonic acid (JA) [167], ethylene (ET) [168], salicylic acid (SA) [169] and abscisic acid (ABA) [45,46,47].

A synergistic effect between GA and H_2_S was observed in seed germination in plants, and this outcome was more evident when treatment with H_2_S was prolonged [170]. It was also observed that GA decreased L-cysteine desulfhydrase (LCD) activity and thus H_2_S production. This enzyme inhibition induced an increase in programmed cell death (PCD) [166]. Nevertheless, exogenous H_2_S treatment alleviated GA-triggered PCD in wheat aleurone cells and blocked the decrease in endogenous H_2_S release by modulating glutathione homeostasis and heme oxygenase-1 gene expression [166].

Auxin is a phytohormone associated with lateral root morphogenesis and root growth regulation. Similar to other phytohormones, a synergistic effect with H_2_S has been observed. Exogenous treatments with H_2_S donors increased the number and length of lateral roots in sweet potato seedlings in a dose-dependent manner [33]. As mentioned previously in this review, crosstalk between the CO/heme oxygenase system and H_2_S is established during lateral root initiation [151]. Furthermore, H_2_S can modulate *CDKA;1*, *CYCA2;1* and Cyclin-dependent kinase inhibitor 2 (*KRP2)* gene expression and act as a downstream component of auxin signaling to activate lateral root formation in tomato [50]. New data shed light on this crosstalk recently, as it was reported that H_2_S inhibited auxin transport through modulation of the subcellular distribution of Peptidyl-prolyl cis-trans isomerase NIMA-interacting (PIN) proteins [171], which is an actin-dependent process. Additionally, it was proven that the regulation of the F-actin cytoskeleton in Arabidopsis roots by H_2_S could affect the auxin distribution in plants [171]. Therefore, the signaling network that includes auxin, H_2_S and F-actin must be finely knotted to regulate root development.

Jasmonic acid regulates diverse plant growth processes and is involved in defense responses against biotic and abiotic stresses. It is well known that H_2_S can regulate abiotic stress tolerance and biotic stress resistance in Arabidopsis [172] and that H_2_S is involved in JA-induced stomatal closure [173]. However, the interaction between H_2_S and JA is still under study. A recent publication demonstrated that treatment with JA promoted endogenous H_2_S generation and that treatments with exogenous H_2_S donors significantly enhanced JA-induced cadmium tolerance [167]. This observation was also described by other authors, whose research described that JA treatments increased D-cysteine desulfhydrase activity and that this JA-induced H_2_S regulated ascorbate and glutathione metabolism [61]. Taken together, these data suggest intertwined signaling between H_2_S and this plant hormone.

Salicylic acid is a phenolic compound involved in local and systemic plant defense responses against pathogens and abiotic stress. SA treatment increased the activity of L-cysteine desulfhydrase and H_2_S accumulation, which improved the heat tolerance of maize seedlings [174]. Contrary to the feedback observed for other hormones and H_2_S, sulfide treatments had no significant effect on SA accumulation and its biosynthesis enzymes [169]. However, a synergistic role was observed between SA and H_2_S in the antioxidant system and osmolyte in crosstalk-induced heat tolerance of maize seedlings [169].

ET is another phytohormone that has been linked with H_2_S signaling. Several authors have described how exogenous treatments with ET induce L- and D-cysteine desulfhydrase activity, and this H_2_S regulates ethylene-induced stomatal closure in *Arabidopsis thaliana* and *Vicia faba* [168,175,176]. A new study revealed that treatments with ET induced H_2_S generation, and feedback regulation was also observed since ethylene-induced H_2_S negatively regulated ethylene biosynthesis; this regulation occurred through the persulfidation of ACO1 in tomato plants [116]. Further investigations have shown that H_2_S may have an antagonistic effect on ethylene, reducing oxidative stress and repressing ethylene synthesis-related gene expression [177].

In recent years, the crosstalk of ABA with H_2_S has been the focus of several investigations since ABA is a key player in plant physiology, mainly under drought stress. It has been broadly reported that H_2_S plays a role in stomatal closure [43,46] and that impaired H_2_S generation mutants (*DES1* knockout Arabidopsis mutants) do not show stomatal closure upon ABA treatment, although this effect could be reversed by exogenous application of H_2_S [46]. This crosstalk was also observed in wheat [178]. Surprisingly, *abi1* mutants were not able to close their stomata in response to sulfide, suggesting that functional ABI1 is required to close the stomata through H_2_S [44]. As described above, H_2_S acts upstream of NO to regulate ABA-induced stomatal closure [44], but H_2_S acts downstream of NO in ethylene-induced stomatal closure [175]. In a parallel study, the authors demonstrated that H_2_S induced ABA-dependent stomatal opening instead [45], which was further demonstrated by Honda et al., who found that H_2_S donors were able to close the stomata during the first 150 min of treatment and induce opening after prolonged treatments [179]. This dual effect could be related to the production of NO in guard cells, and therefore, a complex crosstalk of H_2_S, NO, ET and ABA might regulate stomatal movement depending on environmental stress. A recent study demonstrated the persulfidation of several proteins involved in ABA signaling and ABA biosynthesis, such as SnRK2.2, a key component and activator of ABA signaling; two ABA receptors, pyrabactin resistance receptor 1 (PYR1) and pyrabactin resistance-like receptor (PYL1); the protein phosphatase 2C (HAB2), which is a repressor of ABA signaling; and the nuclear transcription factor Y (NFYC4), which is involved in the ABA signaling pathway [105]. Another enzyme that was shown to be persulfidated in this study was phospholipase D, the activity of which was demonstrated to be regulated by H_2_S to control stomatal closure [180]. Other studies demonstrated the *s*-nitrosation of some proteins involved in ABA signaling, such as the leucine zipper transcription factor Abscisic acid insensitive 5 (ABI5); SnRK2.2, which was also persulfidated; and Open Stomata1 (OST1) [181,182]. The mechanism of action of H_2_S and NO in this tight regulation has been proposed to be through persulfidation and *s*-nitrosation of proteins that play key roles in ABA signaling [105,182]. More recently, the mechanism of action of this crosstalk was deciphered; ABA stimulates the persulfidation of L-cysteine desulfhydrase 1, and H_2_S accumulation drives persulfidation of the NADPH oxidase respiratory burst oxidase homolog protein D (RBOHD), enhancing its activity and triggering stomatal closure [183].

### 5.5. Thioredoxins

As proposed by several authors, one mechanism of action of H_2_S is modulation of protein persulfidation, but the thiol group may undergo a wide range of posttranslational modifications (PTMs) in cells, including oxidation to disulfide (-S-S-), sulfenylation (-SOH), sulfinylation (-SO_2_H), and sulfonylation (-SO_3_H); *s*-nitrosation (-SNO) and glutathionylation (-SS-glutathione). Some of these PTMs can be chemically reversible by reductants in vivo, such as glutathione, or by a cysteine nucleophilic attack to rebuild a disulfide bond. Thioredoxins (TRX) are small oxidoreductases that mainly reduce oxidized cysteines and cleave disulfide bonds. However, TRX may also act as transpersulfidases [184], denitrosylases [185,186] or deglutathionylases [187]. Hence, modified cysteines can be restored to a thiol group. In that sense, persulfidation may protect cysteine residues from the other oxidative modifications, which can be eventually more damaging or irreversible. Deregulation of H_2_S, NO or glutathione levels in the cell can be devastating, and these signaling molecules can reversibly modify proteins. Thioredoxin could be necessary to restore a native protein and transfer the modification to another protein to fulfil other outcomes.

In a prolonged oxidative environment, thiol oxidation leads to the irreversible formation of sulfinic and sulfonic acids. H_2_S has been proposed to act as a protective molecule to avoid these irreversible modifications, since persulfidated proteins can react with reactive oxygen/nitrogen species but can also be reduced by thioredoxins to restore the thiol group [108]. The role of thioredoxins in maintaining persulfidation has been reported in human embryonic kidney cells and the mouse liver because two thioredoxin knockdown cells showed increased polysulfide and protein persulfidation levels [188]. In a recent study, it has been demonstrated that H_2_S regulates the redox state of Trx, disrupting the H_2_O_2_-initiated polymerization of Trx, modulating this antioxidant system [189] 

The interaction between H_2_S and a wide number of other signaling molecules indicates that H_2_S is an essential molecule of signaling in cell life (Figure 2).

## 6. H_2_S in Human and Plant Therapies

It is well known that sulfurous water baths were used by ancient civilizations and were known to have healing effects against particular diseases. H_2_S has been recognized as having anti-inflammatory, anti-bacterial, vasodilator, and anti-fungal properties owing to its sulfur content [68,190,191]. Several extracts from the genus Allium, mainly onion and garlic, and their derivatives have been used since ancient times in China as medicines to treat numerous diseases, including cancer [192], cardiovascular disease [193] and other diseases. It is known that these extracts are a source of sulfur-containing flavor compounds such as diallyl sulfide, allicin and cycloalliin, among others, and which release H_2_S in cells upon interaction with reductants [194,195].

Currently, these beneficial effects are still under study to develop new strategies and therapies to treat certain diseases in mammals and to address agricultural challenges. In mammals, therapies that include H_2_S are used for their anti-inflammatory effects, cytoprotective properties and antiapoptotic features [196]. The aim of these therapies is to be able to use this signaling molecule in heart failure, neurodegenerative diseases and stroke, and ischemia, among others. There has recently been an increasing number of publications indicating that deficiency or excess sulfur amino acids (SAAs), namely, methionine and cysteine, in the diet affect the normal growth of animals and that it is important that SAAs are ingested at the appropriate dose [197,198], since they affect signaling in cells through H_2_S [199]. These amino acids are metabolized through the transsulfuration pathway, which is the one of the main sources of H_2_S in cells; H_2_S has been shown to increase the lifespan of *C. elegans* [200] and even humans [201].

Nevertheless, clinical research on H_2_S is not easy to perform due to its toxicity, and H_2_S therapy is still in a preliminary preclinical stage. A bottleneck for developing gasotransmitter-based therapeutics is the lack of a safe administration drug. There are some candidate compounds for CO and NO prodrugs [202,203,204] and more interestingly, some H_2_S-releasing drugs are currently in clinical trials [205,206]. In a recent study, intraperitoneal injections of JK-1 (a H_2_S donor) were administered to mice after transverse aortic constriction and were shown to have substantial beneficial effects on renal and vascular function [207]. Another exciting approach was a high increase in the dietary intake of taurine, which boosted CSE-mediated H_2_S production to exert significant protective effects in atherogenesis, hypertension and heart failure [208]. However, most therapies use an increase in the dietary intake of sulfur amino acids or directly use slow-releasing H_2_S donors to avoid the toxicity of high H_2_S concentrations [209]. These therapies in mouse models can be used as models to study H_2_S donors in humans. A recent study revealed that persulfidation decreases with aging and that dietary/pharmacological interventions could be used to increase persulfidation and extend lifespan [210]. Moreover, a few recently published articles described the interplay between H_2_S, CO and NO within the gastrointestinal tract, especially in ulcer healing and prevention of non-steroidal anti-inflammatory drugs (NSAIDs)—induced gastropathy [211,212]. In addition, a novel H_2_S donor not only increases H_2_S levels, but also increases circulating NO bioavailability in heart failure patients, highlighting the crosstalk between these gasotransmitters in therapeutic trials [213]. 

In plants, new therapies or strategies using H_2_S are being used to deal with economic losses due to fruit and vegetable ripening or crop stress resistance. It has been shown that H_2_S fumigation slows fruit ripening and senescence in fruits and vegetables by inducing antioxidant activities, such as ascorbate peroxidase, catalase, peroxidase, glutathione reductase, and superoxide dismutase [214,215,216]. Treatments with exogenous H_2_S have also been used to control the color degradation of certain horticultural vegetables and fruits by suppressing the degradation of anthocyanins [214] and downregulating some chlorophyll degradation genes [217]. Interactions between H_2_S and other signaling molecules, such as NO and ethylene, have also been a focus of recent investigations on the senescence of flowers or ripening of fruits. Hydrogen sulfide alleviates postharvest ripening and senescence of fruits by antagonizing the effect of ethylene [218,219]. In addition, a cooperative effect of H_2_S and NO has been observed on delaying the softening and decay of fruits [220], and the crosstalk between these two gasotransmitters is associated with the inhibition of ethylene biosynthesis [221].

There is a long list of publications on the beneficial effects of H_2_S treatments in crops, such as enhancing resistance to metal, heat, cold, salt and drought stresses, which have been recently summarized [222]. It has been demonstrated that sulfur fertilization of crops reduces sensitivity to pathogens, in a process mediated by hydrogen sulfide [16]. H_2_S-induced pathogen resistance is conferred through increased expression of salicylic acid-dependent pathogen-related (PR) genes [223] and increased transcription levels of microRNA393 (*MIR393*) genes [39]. Another important beneficial effect of H_2_S treatment of crops is its influence on the modulation of photosynthesis [34] and autophagy regulation [57]. Apparently, H_2_S is able to regulate energy production in mitochondria, protecting against aging and increasing the lifespan of plants in a similar way as in animals [224]. All of these advantageous outcomes lead to increased yields and biomass and enhanced germination of agricultural crops after H_2_S administration [48,225].

## 7. Conclusions and Future Research

Early life forms had to survive in an atmosphere that contained highly reactive compounds, such as NO, CO and H_2_S, and it seems that during evolution, early life forms not only tolerated these compounds but also included them as important molecules in their signaling mechanisms.

In this review, we summarized the wide promiscuity of H_2_S, which is able to react with a broad range of signaling molecules, acting on its own or in cooperation with those molecules. In addition, we showed that a wide series of pathways are regulated by H_2_S, including either important physiological pathways and pathophysiological or stress conditions. Persulfidation has been proposed to be the mechanism of action of H_2_S in cells throughout all *regna,* but how this modification affects individual proteins and the general consequences on signaling pathways needs further study. The instability of H_2_S and persulfidated cysteines and the imprecise quantitative detection methods for them have slowed the progress of research. Further investigation into developing an appropriate detection method is crucial to understanding H_2_S signaling. Future studies on the compartmentalization or microenvironment of these molecules will be important for studying different modifications on the same target and their biological outcomes. A better understanding of this signaling pathway would shed light on new targets for medical therapies and agricultural remedies.

## Figures and Tables

**Figure 1 antioxidants-09-00621-f001:**
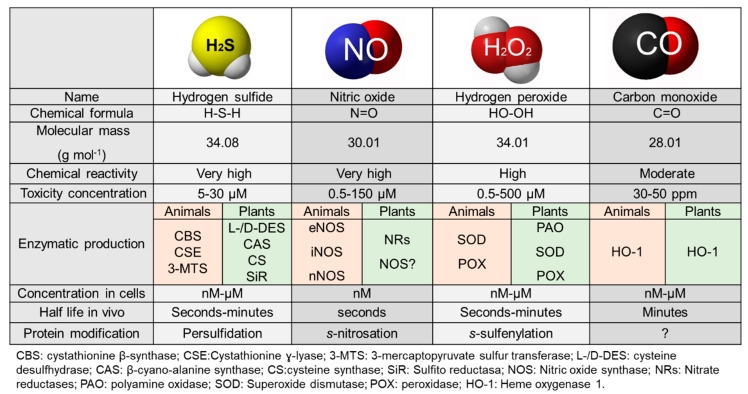
Comparison of signaling molecules in plants and mammals. Hydrogen sulfide (H_2_S), nitric oxide (NO), hydrogen peroxide (H_2_O_2_) and carbon monoxide (CO) are considered to be signaling molecules in diverse and important physiological pathways in cells. These inorganic molecules are endogenously produced by enzymatic pathways and have similar molecular masses but different chemical reactivity.

**Figure 2 antioxidants-09-00621-f002:**
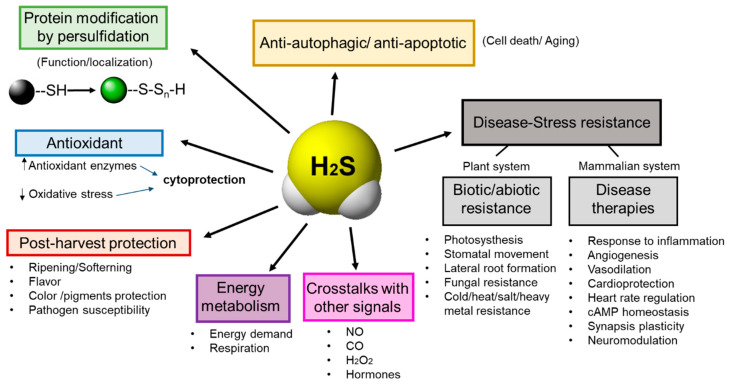
H_2_S is a key activator of multiple physiological processes. H_2_S–mediated signaling ranges from protein modification by persulfidation to affecting a broad range of physiological processes, including regulation of oxidative stress, postharvest protection, disease resistance, autophagy signaling, energy metabolism regulation and crosstalk with other signaling molecules.

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
