# Peer review of "Hydrogen Sulfide: From a Toxic Molecule to a Key Molecule of Cell Life"

_antioxidants, 2020, doi:10.3390/antiox9070621_

Round 1

Reviewer 1 Report

The review article titled " Hydrogen sulfide: From a toxic molecule to a key 2 molecule of cell life" is well written and informative of the role of H2S in biological systems, its regulation, and its cross-talk as a gasotransmitter with other molecules.

Minor corrections

The detection of H2S section 3 is a bit lengthy. As per the abstract, the focus should be more on the cross talk and its regulation in living systems. Either a tabulated form of the detection methods or citation of a review article that has focused on H2S detection can be used.

Line 14 it should be summarized

Line 33 define HCN

Lines 118,128, 184, and 418 in vivo should be italicized unless that is the form as required by the journal.

Lines 237 and 238. The paragraph starting on Line 238 seems to have a slight disconnect from the previous paragraph.

Line 243 define MgrA

Line 352 define KRP2

Line 355 define PIN

Line 376 Ethylene (ET) should be abbreviated here as it is mentioned as ET elsewhere.

Line 381 persulfidation of ACO. What is ACO?

Line 399 define PYR1 and PYL1

Lines 405 and 406 remove the capitalization of Abscisic Acid Insensitive 5 and Open Stomata 1

Line 410 define RBOHD

Line 482 what is SA-dependent PR genes?

Line 482 define MIR393

Author Response

Dear reviewer,

thanks so much for your comments. A detailed information about the changes made on the text, based on your suggestions, is provided:

Minor corrections

The detection of H2S section 3 is a bit lengthy. As per the abstract, the focus should be more on the cross talk and its regulation in living systems. Either a tabulated form of the detection methods or citation of a review article that has focused on H2S detection can be used.

Answer: Thanks for the comments. A tabulated form of detection methods has been updated and crosstalk among H2S and other signaling molecules has been extended.

Line 14 it should be summarized.

Answer: it has been summarized and highlighted in yellow

Line 33 define HCN

Lines 118,128, 184, and 418 in vivo should be italicized unless that is the form as required by the journal.

Line 243 define MgrA

Line 352 define KRP2

Line 355 define PIN

Line 376 Ethylene (ET) should be abbreviated here as it is mentioned as ET elsewhere.

Line 399 define PYR1 and PYL1

Lines 405 and 406 remove the capitalization of Abscisic Acid Insensitive 5 and Open Stomata 1

Line 410 define RBOHD

Line 482 define MIR393

Answer: all these suggestions have been defined in the text and highlighted in yellow. Thanks for the corrections, those changes will make this article more understandable for unrelated readers

Line 381 persulfidation of ACO. What is ACO?

Answer: ACO is ACO1, which is 1-aminocyclopropane-1-carboxylic acid oxidase and described in line 237. Line 381 had a typo which has been corrected.

Lines 237 and 238. The paragraph starting on Line 238 seems to have a slight disconnect from the previous paragraph.

Answer: that paragraph has been rewritten to make it fit better and more understandable. Now it is in lines 240-243.

Line 482 what is SA-dependent PR genes?

Answer: salicylic acid-dependent pathogen related genes. It has been updated in the text and highlighted in yellow.

Reviewer 2 Report

This manuscript is interesting and provides new knowledge related to gaseous mediators in pathophysiology and physiology. I would recommend to revise the section addressing possible crosstalk between these molecules. It is worth to cite few recently published articles in the field of pharmacology and molecular biology regarding interaction and interplay between H2S, CO and/or NO within gastrointestinal tract, including especially ulcer healing and prevention of drugs (especially NSAIDS including aspirin)-induced gastropathy. Moreover, it is also important to mention that H2S-releasing drugs are currently on clinical trials (please, see e.g. Wallace et al. doi: 10.1111/bph.14641). Also it would be important to add few information and references mentioning at least briefly chemical donors of H2S, CO and NO (e.g. recently published doi: 10.1021/acs.jmedchem.9b00073).

Author Response

Answer: Thanks so much for your comments. The section addressing crosstalks between these molecules has been updated and extended. Some new recent references have been included. It was worthy to include all information and references suggested by the reviewer. These improvements of the text have been highlighted in yellow in lines 473-490.

Reviewer 3 Report

Overall, this is an interesting review that hits the important points related to the intended topic of coverage. A few suggested modifications to improve the scope and generality are included below.
1. The statement that ‘several articles published more recently showed the beneficial properties” is misleading to a reader who is not familiar with the area since there are likely 1000+ articles that fit this description rather than ‘several’
2. In Section 3: It is worth noting the different speciation of the H2S, HS-, S(2-), since there is virtually no S(2-) at normal pHs.
3. In the discussion of the methylene blue method, is stated that it has a sensitivity of 3-500 micrograms, but since this is a solution measurement this should be reported in concentration (since that mass range is very different if the volumes are very different). Also in discussion, the symbol used for the ‘p’ in N,N-dimethyl-p-phenylenediamine is incorrect.
4. In the discussion of the first measurements of biological H2S at 5—160 micromolar, the authors should reinforce in this review that these early numbers are incorrect. (as a clear example, the human nose can detect 1 micromolar H2S in buffer!)
5. In the discussion of the monobromobimane method, it should also be stated that this method allows for separation and quantification of the ‘free’ and ‘sulfane-sulfur’ H2S pools.
6. In the discussion of the chemistry of H2S / NO crosstalk. A number of other references that point to the underlying chemistry at action should be included. As examples: https://doi.org/10.1021/acs.inorgchem.9b01978 ; https://doi.org/10.1021/acs.inorgchem.6b01660 ; https://doi.org/10.1039/C6CP06314D
7. The discussion of H2S-related therapies focuses only on selected animal models. The coverage in this section seems unbalanced and may be better providing a brief summary of ongoing clinical trial work on H2S-releasing compounds, which would match the topic of H2S therapies more closely.
8. Ref 95 is missing page and volume numbers.

Author Response

Dear Reviewer, thanks so much for your comments and suggestions. A detailed information about the changes done to the manuscript following your instructions, is described in the following sections. 

  1. The statement that ‘several articles published more recently showed the beneficial properties” is misleading to a reader who is not familiar with the area since there are likely 1000+ articles that fit this description rather than ‘several’

Answer: Thanks for the comment. You are completely right. It was confusing and that sentence has been changed in line 9.

  1. In Section 3: It is worth noting the different speciation of the H2S, HS-, S(2-), since there is virtually no S(2-) at normal pHs.

Answer: The suggestion has been included in the text as ` H2S in aqueous solution can be found as hydrogen sulfide gas (H2S) or in one of its dissociated forms, hydrosulfide anions (HS) and sulfide anions (S2−), although at physiological pH, S2- is only found in a negligible concentration.´

  1. In the discussion of the methylene blue method, is stated that it has a sensitivity of 3-500 micrograms, but since this is a solution measurement this should be reported in concentration (since that mass range is very different if the volumes are very different). Also in discussion, the symbol used for the ‘p’ in N,N-dimethyl-p-phenylenediamine is incorrect.

Answer: the symbol has been changed. The concentration has not been updated since the reference from 1949 did not reveal the concentration, but sentence has been corrected and exact concentrations have been provided for the improved methylene blue method. Lines 122-125

  1. In the discussion of the first measurements of biological H2S at 5—160 micromolar, the authors should reinforce in this review that these early numbers are incorrect. (as a clear example, the human nose can detect 1 micromolar H2S in buffer!)

Answer: we appreciate the comment. We corrected and added some other references. Lines 179-184.

  1. In the discussion of the monobromobimane method, it should also be stated that this method allows for separation and quantification of the ‘free’ and ‘sulfane-sulfur’ H2S pools.

Answer: the text has been corrected including this suggestion. Line 134

  1. In the discussion of the chemistry of H2S / NO crosstalk. A number of other references that point to the underlying chemistry at action should be included. As examples: https://doi.org/10.1021/acs.inorgchem.9b01978 ; https://doi.org/10.1021/acs.inorgchem.6b01660 ; https://doi.org/10.1039/C6CP06314D

Answer: The discussion of H2s/NO crosstalk has been updated including these references suggested by the reviewer. They have been cited in lines 290-293.

  1. The discussion of H2S-related therapies focuses only on selected animal models. The coverage in this section seems unbalanced and may be better providing a brief summary of ongoing clinical trial work on H2S-releasing compounds, which would match the topic of H2S therapies more closely.

Answer: The discussion has been updated including some references of clinical trials with H2S releasing compounds and CO/NO prodrugs. Lines 473-490.

  1. Ref 95 is missing page and volume numbers.

Answer: Thanks for letting us know. It has been corrected. Ref is now number 106

Round 2

Reviewer 2 Report

Thank you very much for adding new references and more details into the text. I would like to kindly ask for a little update. Authors stated in revised MS about interplay between the gaseous mediators mentioning ulcer healing but there is no appropriate reference added related directly to ulcer healing, I suggest e.g.: doi: 10.1096/fj.07-8669com; doi: 10.1111/bph.13968. 

Author Response

Thanks so much for your suggestion. I have modified the references regarding the interplay between the gaseous mediators in ulcer healing, as suggested by the reviewer.